# Variational Inference with Tail-adaptive $f$-Divergence

Dilin Wang
UT Austin
dilin@cs.utexas.edu

Hao Liu *
UESTC
uestcliuhao@gmail.com

Qiang Liu
UT Austin
lqiang@cs.utexas.edu

## Abstract

Variational inference with $\alpha$-divergences has been widely used in modern probabilistic machine learning. Compared to Kullback-Leibler (KL) divergence, a major advantage of using $\alpha$-divergences (with positive $\alpha$ values) is their *mass-covering* property. However, estimating and optimizing $\alpha$-divergences require to use importance sampling, which may have large or infinite variance due to heavy tails of importance weights. In this paper, we propose a new class of *tail-adaptive $f$-divergences* that adaptively change the convex function $f$ with the tail distribution of the importance weights, in a way that theoretically guarantees finite moments, while simultaneously achieving mass-covering properties. We test our method on Bayesian neural networks, and apply it to improve a recent soft actor-critic (SAC) algorithm (Haarnoja et al., 2018) in deep reinforcement learning. Our results show that our approach yields significant advantages compared with existing methods based on classical KL and $\alpha$-divergences.

## 1 Introduction

Variational inference (VI) (e.g., Jordan et al., 1999; Wainwright et al., 2008) has been established as a powerful tool in modern probabilistic machine learning for approximating intractable posterior distributions. The basic idea is to turn the approximation problem into an optimization problem, which finds the best approximation of an intractable distribution from a family of tractable distributions by minimizing a divergence objective function. Compared with Markov chain Monte Carlo (MCMC), which is known to be consistent but suffers from slow convergence, VI provides biased results but is often practically faster. Combined with techniques like stochastic optimization (Ranganath et al., 2014; Hoffman et al., 2013) and reparameterization trick (Kingma & Welling, 2014), VI has become a major technical approach for advancing Bayesian deep learning, deep generative models and deep reinforcement learning (e.g., Kingma & Welling, 2014; Gal & Ghahramani, 2016; Levine, 2018).

A key component of successful variational inference lies on choosing a proper divergence metric. Typically, closeness is defined by the KL divergence $\text{KL}(q \, || \, p)$ (e.g., Jordan et al., 1999), where $p$ is the intractable distribution of interest and $q$ is a simpler distribution constructed to approximate $p$. However, VI with KL divergence often under-estimates the variance and may miss important local modes of the true posterior (e.g., Christopher, 2016; Blei et al., 2017). To mitigate this issue, alternative metrics have been studied in the literature, a large portion of which are special cases of $f$-divergence (e.g., Csiszár & Shields, 2004):

$$D_f(p \, || \, q) = \mathbb{E}_{x \sim q} \left[ f\left( \frac{p(x)}{q(x)} \right) - f(1) \right], \tag{1}$$

where $f : \mathbb{R}_+ \rightarrow \mathbb{R}$ is any convex function. The most notable class of $f$-divergence that has been exploited in VI is $\alpha$-divergence, which takes $f(t) = t^\alpha/(\alpha(\alpha - 1))$ for $\alpha \in \mathbb{R} \setminus \{0, 1\}$. By choosing different $\alpha$, we get a large number of well-known divergences as special cases, including the standard

KL divergence objective $\mathrm{KL}(q \,||\, p)$ ($\alpha \to 0$), the KL divergence with the reverse direction $\mathrm{KL}(p \,||\, q)$ ($\alpha \to 1$) and the $\chi^2$ divergence ($\alpha = 2$). In particular, the use of general $\alpha$-divergence in VI has been widely discussed (e.g., Minka et al., 2005; Hernández-Lobato et al., 2016; Li & Turner, 2016); the reverse KL divergence is used in expectation propagation (Minka, 2001; Opper & Winther, 2005), importance weighted auto-encoders (Burda et al., 2016), and the cross entropy method (De Boer et al., 2005); $\chi^2$-divergence is exploited for VI (e.g., Dieng et al., 2017), but is more extensively studied in the context of adaptive importance sampling (IS) (e.g., Cappé et al., 2008; Ryu & Boyd, 2014; Cotter et al., 2015), since it coincides with the variance of the IS estimator with $q$ as the proposal.

A major motivation of using $\alpha$-divergence contributes to its *mass-covering* property: when $\alpha > 0$, the optimal approximation $q$ tends to cover more modes of $p$, and hence better accounts for the uncertainty in $p$. Typically, larger values of $\alpha$ enforce stronger mass-covering properties. In practice, however, $\alpha$ divergence and its gradient need to be estimated empirically using samples from $q$. Using large $\alpha$ values may cause high or infinite variance in the estimation because it involves estimating the $\alpha$-th power of the density ratio $p(x)/q(x)$, which is likely distributed with a heavy or fat tail (e.g., Resnick, 2007). In fact, when $q$ is very different from $p$, the expectation of ratio $(p(x)/q(x))^\alpha$ can be infinite (that is, $\alpha$-divergence does not exist). This makes it problematic to use large $\alpha$ values, despite the mass-covering property it promises. In addition, it is reasonable to expect that the optimal setting of $\alpha$ should vary across training processes and learning tasks. Therefore, it is desirable to design an approach to choose $\alpha$ *adaptively* and *automatically* as $q$ changes during the training iterations, according to the distribution of the ratio $p(x)/q(x)$.

Based on theoretical observations on $f$-divergence and fat-tailed distributions, we design a new class of $f$-divergence which is *tail-adaptive* in that it uses different $f$ functions according to the tail distribution of the density ratio $p(x)/q(x)$ to simultaneously obtain stable empirical estimation and a strongest possible mass-covering property. This allows us to derive a new adaptive $f$-divergence-based variational inference by combining it with stochastic optimization and reparameterization gradient estimates. Our main method (Algorithm 1) has a simple form, which replaces the $f$ function in (1) with a rank-based function of the empirical density ratio $w = p(x)/q(x)$ at each gradient descent step of $q$, whose variation depends on the distribution of $w$ and does not explode regardless the tail of $w$.

Empirically, we show that our method can better recover multiple modes for variational inference. In addition, we apply our method to improve a recent soft actor-critic (SAC) algorithm (Haarnoja et al., 2018) in reinforcement learning (RL), showing that our method can be used to optimize multi-modal loss functions in RL more efficiently.

## 2 $f$-Divergence and Friends

Given a distribution $p(x)$ of interest, we want to approximate it with a simpler distribution from a family $\{q_\theta(x) \colon \theta \in \Theta\}$, where $\theta$ is the variational parameter that we want to optimize. We approach this problem by minimizing the $f$-divergence between $q_\theta$ and $p$:

$$\min_{\theta \in \Theta} \left\{ D_f(p \,||\, q_\theta) = \mathbb{E}_{x \sim q_\theta} \left[ f\left( \frac{p(x)}{q_\theta(x)} \right) - f(1) \right], \right\} \tag{2}$$

where $f \colon \mathbb{R}_+ \to \mathbb{R}$ is any twice differentiable convex function. It can be shown by Jensen's inequality that $\mathbb{D}_f(p \,||\, q) \geq 0$ for any $p$ and $q$. Further, if $f(t)$ is strictly convex at $t = 1$, then $D_f(p \,||\, q) = 0$ implies $p = q$. The optimization in (2) can be solved approximately using stochastic optimization in practice by approximating the expectation $\mathbb{E}_{x \sim q_\theta}[\cdot]$ using samples drawing from $q_\theta$ at each iteration.

The $f$-divergence includes a large spectrum of important divergence measures. It includes KL divergence in both directions,

$$\mathrm{KL}(q \,||\, p) = \mathbb{E}_{x \sim q} \left[ \log \frac{q(x)}{p(x)} \right], \qquad \mathrm{KL}(p \,||\, q) = \mathbb{E}_{x \sim q} \left[ \frac{p(x)}{q(x)} \log \frac{p(x)}{q(x)} \right], \tag{3}$$

which correspond to $f(t) = -\log t$ and $f(t) = t \log t$, respectively. $\mathrm{KL}(q \,||\, p)$ is the typical objective function used in variational inference; the reversed direction $\mathrm{KL}(p \,||\, q)$ is also used in various settings (e.g., Minka, 2001; Opper & Winther, 2005; De Boer et al., 2005; Burda et al., 2016).

More generally, $f$-divergence includes the class of $\alpha$-divergence, which takes $f_\alpha(t) = t^\alpha/(\alpha(\alpha-1))$, $\alpha \in \mathbb{R} \setminus \{0, 1\}$ and hence

$$D_{f_\alpha}(p \,||\, q) = \frac{1}{\alpha(\alpha-1)} \mathbb{E}_{x \sim q} \left[ \left( \frac{p(x)}{q(x)} \right)^\alpha - 1 \right]. \tag{4}$$

One can show that $\mathrm{KL}(q \,||\, p)$ and $\mathrm{KL}(p \,||\, q)$ are the limits of $D_{f_\alpha}(q \,||\, p)$ when $\alpha \to 0$ and $\alpha \to 1$, respectively. Further, one obtain Helinger distance and $\chi^2$-divergence as $\alpha = 1/2$ and $\alpha = 2$, respectively. In particular, $\chi^2$-divergence ($\alpha = 2$) plays an important role in adaptive importance sampling, because it equals the variance of the importance weight $w = p(x)/q(x)$ and minimizing $\chi^2$-divergence corresponds to finding an optimal importance sampling proposal.

## 3  $\alpha$-Divergence and Fat Tails

A major motivation of using $\alpha$ divergences as the objective function for approximate inference is their *mass-covering* property (also known as the zero-avoiding behavior). This is because $\alpha$-divergence is proportional to the $\alpha$-th moment of the density ratio $p(x)/q(x)$. When $\alpha$ is positive and large, large values of $p(x)/q(x)$ are strongly penalized, preventing the case of $q(x) \ll p(x)$. In fact, whenever $D_{f_\alpha}(p \,||\, q) < \infty$, we have $p(x) > 0$ imply $q(x) > 0$. This means that the probability mass and local modes of $p$ are taken into account in $q$ properly.

Note that the case when $\alpha \leq 0$ exhibits the opposite property, that is, $p(x) = 0$ must imply $q(x) = 0$ to make $D_{f_\alpha}(q||p)$ finite when $\alpha \leq 0$; this includes the typical KL divergence $\mathrm{KL}(q \,||\, p)$ ($\alpha = 0$), which is often criticized for its tendency to under-estimate the uncertainty.

Typically, using larger values of $\alpha$ enforces stronger *mass-covering* properties. In practice, however, larger values of $\alpha$ also increase the variance of the empirical estimators, making it highly challenging to optimize. In fact, the expectation in (4) may not even exist when $\alpha$ is too large. This is because the density ratio $w := p(x)/q(x)$ often has a fat-tailed distribution.

A non-negative random variable $w$ is called fat-tailed[2] (e.g., Resnick, 2007) if its tail probability $\bar{F}_w(t) := \Pr(w \geq t)$ is asymptotically equivalent to $t^{-\alpha_*}$ as $t \to +\infty$ for some finite positive number $\alpha_*$ (denoted by $\bar{F}_w(t) \sim t^{-\alpha_*}$), which means that

$$\bar{F}_w(t) = t^{-\alpha_*} L(t),$$

where $L$ is a slowly varying function that satisfies $\lim_{t \to +\infty} L(ct)/L(t) = 1$ for any $c > 0$. Here $\alpha_*$ determines the fatness of the tail and is called the tail index of $w$. For a fat-tailed distribution with index $\alpha_*$, its $\alpha$-th moment exists only if $\alpha < \alpha_*$, that is, $\mathbb{E}[w^\alpha] < \infty$ iff $\alpha < \alpha_*$. It turns out the density ratio $w := p(x)/q(x)$, when $x \sim q$, tends to have a fat-tailed distribution when $q$ is more peaked than $p$. The example below illustrates this with simple Gaussian distributions.

**Example 3.1.** *Assume $p(x) = \mathcal{N}(x; 0, \sigma_p^2)$ and $q(x) = \mathcal{N}(x; 0, \sigma_q^2)$. Let $x \sim q$ and $w = p(x)/q(x)$ the density ratio. If $\sigma_p > \sigma_q$, then $w$ has a fat-tailed distribution with index $\alpha_* = \sigma_p^2/(\sigma_p^2 - \sigma_q^2)$.*

*On the other hand, if $\sigma_p \leq \sigma_q$, then $w$ is bounded and not fat-tailed (effectively, $\alpha_* = +\infty$).*

By the definition above, if the importance weight $w = p(x)/q(x)$ has a tail index $\alpha_*$, the $\alpha$-divergence $D_{f_\alpha}(p \,||\, q)$ exists only if $\alpha < \alpha_*$. Although it is desirable to use $\alpha$-divergence with large values of $\alpha$ as VI objective function, it is important to keep $\alpha$ smaller than $\alpha_*$ to ensure that the objective and gradient are well defined. The problem, however, is that the tail index $\alpha_*$ is unknown in practice, and may change dramatically (e.g., even from finite to infinite) as $q$ is updated during the optimization process. This makes it suboptimal to use a pre-fixed $\alpha$ value. One potential way to address this problem is to estimate the tail index $\alpha^*$ empirically at each iteration using a tail index estimator (e.g., Hill et al., 1975; Vehtari et al., 2015). Unfortunately, tail index estimation is often challenging and requires a large number of samples. The algorithm may become unstable if $\alpha_*$ is over-estimated.

## 4  Hessian-based Representation of $f$-Divergence

In this work, we address the aforementioned problem by designing a generalization of $f$-divergence in which $f$ adaptively changes with $p$ and $q$, in a way that always guarantees the existence of the

expectation, while simultaneous achieving (theoretically) strong mass-covering equivalent to that of the $\alpha$-divergence with $\alpha = \alpha^*$.

One challenge of designing such adaptive $f$ is that the convex constraint over function $f$ is difficult to express computationally. Our first key observation is that it is easier to specify a convex function $f$ through its second order derivative $f''$, which can be any non-negative function. It turns out $f$-divergence, as well as its gradient, can be conveniently expressed using $f''$, without explicitly defining the original $f$.

**Proposition 4.1.** *1) Any twice differentiable convex function $f : \mathbb{R}_+ \cup \{0\} \to \mathbb{R}$ with finite $f(0)$ can be decomposed into linear and nonlinear components as follows*

$$f(t) = (at + b) + \int_0^\infty (t - \mu)_+ h(\mu) d\mu, \tag{5}$$

*where $h$ is a non-negative function, $(t)_+ = \max(0, t)$, and $a, b \in \mathbb{R}$. In this case, $h = f''(t)$, $a = f'(0)$ and $b = f(0)$. Conversely, any non-negative function $h$ and $a, b \in \mathbb{R}$ specifies a convex function.*

*2) This allows us to derive an alternative representation of $f$-divergence:*

$$D_f(p \,||\, q) = \int_0^\infty f''(\mu) \mathbb{E}_{x \sim q} \left[ \left( \frac{p(x)}{q(x)} - \mu \right)_+ \right] d\mu - c, \tag{6}$$

*where $c := \int_0^1 f''(\mu)(1 - \mu) d\mu = f(1) - f(0) - f'(0)$ is a constant.*

*Proof.* If $f(t) = (at + b) + \int_0^\infty (t - \mu)_+ h(\mu) d\mu$, calculation shows

$$f'(t) = a + \int_0^t h(\mu) d\mu, \quad f''(t) = h(t).$$

Therefore, $f$ is convex iff $h$ is non-negative. See Appendix for the complete proof. $\qquad \square$

Eq (6) suggests that all $f$-divergences are conical combinations of a set of special $f$-divergences of form $\mathbb{E}_{x \sim q}[(p(x)/q(x) - \mu)_+ - f(1)]$ with $f(t) = (t - \mu)_+$. Also, every $f$-divergence is completely specified by the Hessian $f''$, meaning that adding $f$ with any linear function $at + b$ does not change $D_f(p \,||\, q)$. Such integral representation of $f$-divergence is not new; see e.g., Feldman & Osterreicher (1989); Osterreicher (2003); Liese & Vajda (2006); Reid & Williamson (2011); Sason (2018).

For the purpose of minimizing $\mathbb{D}_f(p \,||\, q_\theta)$ ($\theta \in \Theta$) in variational inference, we are more concerned with calculating the gradient, rather than the $f$-divergence itself. It turns out the gradient of $\mathbb{D}_f(p \,||\, q_\theta)$ is also directly related to Hessian $f''$ in a simple way.

**Proposition 4.2.** *1) Assume $\log q_\theta(x)$ is differentiable w.r.t. $\theta$, and $f$ is a differentiable convex function. For $f$-divergence defined in (2), we have*

$$\nabla_\theta D_f(p \,||\, q_\theta) = -\mathbb{E}_{x \sim q_\theta} \left[ \rho_f \left( \frac{p(x)}{q_\theta(x)} \right) \nabla_\theta \log q_\theta(x) \right], \tag{7}$$

*where $\rho_f(t) = f'(t)t - f(t)$ (equivalently, $\rho_f'(t) = f''(t)t$ if $f$ is twice differentiable).*

*2) Assume $x \sim q_\theta$ is generated by $x = g_\theta(\xi)$ where $\xi \sim q_0$ is a random seed and $g_\theta$ is a function that is differentiable w.r.t. $\theta$. Assume $f$ is twice differentiable and $\nabla_x \log(p(x)/q_\theta(x))$ exists. We have*

$$\nabla_\theta D_f(p \,||\, q_\theta) = -\mathbb{E}_{x = g_\theta(\xi), \xi \sim q_0} \left[ \gamma_f \left( \frac{p(x)}{q_\theta(x)} \right) \nabla_\theta g_\theta(\xi) \nabla_x \log(p(x)/q_\theta(x)) \right], \tag{8}$$

*where $\gamma_f(t) = \rho_f'(t)t = f''(t)t^2$.*

The result above shows that the gradient of $f$-divergence depends on $f$ through $\rho_f$ or $\gamma_f$. Taking $\alpha$-divergence ($\alpha \notin \{0, 1\}$) as example, we have

$$f(t) = t^\alpha / (\alpha(\alpha - 1)), \qquad \rho_f(t) = t^\alpha / \alpha, \qquad \gamma_f(t) = t^\alpha,$$

all of which are proportional to the power function $t^\alpha$. For KL$(q \parallel p)$, we have $f(t) = -\log t$, yielding $\rho_f(t) = \log t - 1$ and $\gamma_f(t) = 1$; for KL$(p \parallel q)$, we have $f(t) = t \log t$, yielding $\rho_f(t) = t$ and $\gamma_f(t) = t$.

The formulas in (7) and (8) are called the *score-function gradient* and *reparameterization gradient* (Kingma & Welling, 2014), respectively. Both equal the gradient in expectation, but are computationally different and yield empirical estimators with different variances. In particular, the score-function gradient in (7) is "gradient-free" in that it does not require calculating the gradient of the distribution $p(x)$ of interest, while (8) is "gradient-based" in that it involves $\nabla_x \log p(x)$. It has been shown that optimizing with reparameterization gradients tend to give better empirical results because it leverages the gradient information $\nabla_x \log p(x)$, and yields a lower variance estimator for the gradient (e.g., Kingma & Welling, 2014).

Our key observation is that we can directly specify $f$ through any increasing function $\rho_f$, or non-negative function $\gamma_f$ in the gradient estimators, without explicitly defining $f$.

**Proposition 4.3.** *Assume $f \colon \mathbb{R}_+ \to \mathbb{R}$ is convex and twice differentiable, then*

*1) $\rho_f$ in (7) is a monotonically increasing function on $\mathbb{R}_+$. In addition, for any differentiable increasing function $\rho$, there exists a convex function $f$ such that $\rho_f = \rho$;*

*2) $\gamma_f$ in (8) is non-negative on $\mathbb{R}_+$, that is, $\gamma_f(t) \geq 0$, $\forall t \in \mathbb{R}_+$. In addition, for any non-negative function $\gamma$, there exists a convex function $f$ such that $\gamma_f = \gamma$;*

*3) if $\rho_f'(t)$ is strictly increasing at $t = 1$ (i.e., $\rho_f'(1) > 0$), or $\gamma_f(t)$ is strictly positive at $t = 1$ (i.e., $\gamma_f(1) > 0$), then $\mathbb{D}_f(p \parallel q) = 0$ implies $p = q$.*

*Proof.* Because $f$ is convex ($f''(t) \geq 0$), we have $\gamma_f(t) = f''(t)t^2 \geq 0$ and $\rho_f'(t) = f''(t)t \geq 0$ on $t \in \mathbb{R}_+$, that is, $\gamma_f$ is non-negative and $\rho_f$ is increasing on $\mathbb{R}_+$. If $\rho_t$ is strictly increasing (or $\gamma_f$ is strictly positive) at $t = 1$, we have $f$ is strictly convex at $t = 1$, which guarantees $\mathbb{D}_f(p \parallel q) = 0$ imply $p = q$.

For non-negative function $\gamma(t)$ (or increasing function $\rho(t)$) on $\mathbb{R}_+$, any convex function $f$ whose second-order derivative equals $\gamma(t)/t^2$ (or $\rho_f'(t)/t$) satisfies $\gamma_f = \gamma$ (resp. $\rho_f = \rho$). □

## 5 Safe $f$-Divergence with Inverse Tail Probability

The results above show that it is sufficient to find an increasing function $\rho_f$, or a non-negative function $\gamma_f$ to obtain adaptive $f$-divergence with computable gradients. In order to make the $f$-divergence "safe", we need to find $\rho_f$ or $\gamma_f$ that adaptively depends on $p$ and $q$ such that the expectation in (7) and (8) always exists. Because the magnitude of $\nabla_\theta \log q_\theta(x)$, $\nabla_x \log(p(x)/q_\theta(x))$ and $\nabla_\theta g_\theta(\xi)$ are relatively small compared with the ratio $p(x)/q(x)$, we can mainly consider designing function $\rho$ (or $\gamma$) such that they yield finite expectation $\mathbb{E}_{x \sim q}[\rho(p(x)/q(x))] < \infty$; meanwhile, we should also keep the function large, preferably with the same magnitude as $t^{\alpha_*}$, to provide a strong mode-covering property. As it turns out, the inverse of the tail probability naturally achieves all these goals.

**Proposition 5.1.** *For any random variable $w$ with tail distribution $\bar{F}_w(t) := \Pr(w \geq t)$ and tail index $\alpha_*$, we have*

$$\mathbb{E}[\bar{F}_w(w)^\beta] < \infty, \qquad \text{for any } \beta > -1.$$

*Also, we have $\bar{F}_w(t)^\beta \sim t^{-\beta \alpha_*}$, and $\bar{F}_w(t)^\beta$ is always non-negative and monotonically increasing when $\beta < 0$.*

*Proof.* Simply note that $\mathbb{E}[\bar{F}_w(w)^\beta] = \int \bar{F}_w(t)^\beta d\bar{F}_\beta(t) = \int_0^1 t^\beta dt$, which is finite only when $\beta > -1$. The non-negativity and monotonicity of $\bar{F}_w(t)^\beta$ are obvious. $\bar{F}_w(t)^\beta \sim t^{-\beta \alpha_*}$ directly follows the definition of the tail index. □

This motivates us to use $\bar{F}_w(t)^\beta$ to define $\rho_f$ or $\gamma_f$, yielding two versions of "safe" tail-adaptive $f$ divergences. Note that here $f$ is defined implicitly through $\rho_f$ or $\gamma_f$. Although it is possible to derive the corresponding $f$ and $D_f(p \parallel q)$, there is no computational need to do so, since optimizing the objective function only requires calculating the gradient, which is defined by $\rho_f$ or $\gamma_f$.

**Algorithm 1** Variational Inference with Tail-adaptive $f$-Divergence (with Reparameterization Gradient)

---

Goal: Find the best approximation of $p(x)$ from $\{q_\theta : \theta \in \Theta\}$. Assume $x \sim q_\theta$ is generated by $x = g_\theta(\xi)$ where $\xi$ is a random sample from noise distribution $q_0$.

Initialize $\theta$, set an index $\beta$ (e.g., $\beta = -1$).

**for** iteration **do**

   Draw $\{x_i\}_{i=1}^n \sim q_\theta$, generated by $x_i = g_\theta(\xi_i)$.

   Let $w_i = p(x_i)/q_\theta(x_i)$, $\hat{\bar{F}}_w(t) = \sum_{j=1}^n \mathbb{I}(w_j \geq t)/n$, and set $\gamma_i = \hat{\bar{F}}_w(w_i)^\beta$.

   Update $\theta \leftarrow \theta + \epsilon \Delta\theta$, with $\epsilon$ is step size, and

$$\Delta\theta = \frac{1}{z_\gamma} \sum_{i=1}^n \left[ \gamma_i \nabla_\theta g_\theta(\xi_i) \nabla_x \log(p(x_i)/q_\theta(x_i)) \right], \quad \text{where} \quad z_\gamma = \sum_{i=1}^n \gamma_i.$$

**end for**

---

In practice, the explicit form of $\bar{F}_w(t)^\beta$ is unknown. We can approximate it based on empirical data drawn from $q$. Let $\{x_i\}$ be drawn from $q$ and $w_i = p(x_i)/q(x_i)$, then we can approximate the tail probability with $\hat{\bar{F}}_w(t) = \frac{1}{n}\sum_{i=1}^n \mathbb{I}(w_i \geq t)$. Intuitively, this corresponds to assigning each data point a weight according to the rank of its density ratio in the population. Substituting the empirical tail probability into the reparametrization gradient formula in (8) and running a gradient descent with stochastic approximation yields our main algorithm shown in Algorithm 1. The version with the score-function gradient is similar and is shown in Algorithm 2 in the Appendix. Both algorithms can be viewed as minimizing the implicitly constructed adaptive $f$-divergences, but correspond to using different $f$.

Compared with typical VI with reparameterized gradients, our method assigns a weight $\rho_i = \hat{\bar{F}}_w(w_i)^\beta$, which is proportional $\#w_i^\beta$ where $\#w_i$ denotes the rank of data $w_i$ in the population $\{w_i\}$. When taking $-1 < \beta < 0$, this allows us to penalize places with high ratio $p(x)/q(x)$, but avoid to be overly aggressive. In practice, we find that simply taking $\beta = -1$ almost always yields the best empirical performance (despite needing $\beta > -1$ theoretically). By comparison, minimizing the classical $\alpha$-divergence would have a weight of $w_i^\alpha$; if $\alpha$ is too large, the weight of a single data point becomes dominant, making gradient estimate unstable.

## 6 Experiments

In this section, we evaluate our adaptive $f$-divergence with different models. We use reparameterization gradients as default since they have smaller variances (Kingma & Welling, 2014) and normally yield better performance than score function gradients. Our code is available at https://github.com/dilinwang820/adaptive-f-divergence.

### 6.1 Gaussian Mixture Toy Example

We first illustrate the approximation quality of our proposed adaptive $f$-divergence on Gaussian mixture models. In this case, we set our target distribution to be a Gaussian mixture $p(x) = \sum_{i=1}^k \frac{1}{k}\mathcal{N}(x; \nu_i, 1)$, for $x \in \mathbb{R}^d$, where the elements of each mean vector $\nu_i$ is drawn from $\text{uniform}([-s, s])$. Here $s$ can be viewed as controlling the Gaussianity of the target distribution: $p$ reduces to standard Gaussian distribution when $s = 0$ and is increasingly multi-modal when $s$ increases. We fix the number of components to be $k = 10$, and initialize the proposal distribution using $q(x) = \sum_{i=1}^{20} w_i \mathcal{N}(x; \mu_i, \sigma_i^2)$, where $\sum_{i=1}^{20} w_i = 1$.

We evaluate the mode-seeking ability of how $q$ covers the modes of $p$ using a "mode-shift distance" $dist(p, q) := \sum_{i=1}^{10} \min_j ||\nu_i - \mu_j||_2 / 10$, which is the average distance of each mode in $p$ to its nearest mode in distribution $q$. The model is optimized using Adagrad with a constant learning rate $0.05$. We use a minibatch of size 256 to approximate the gradient in each iteration. We train the model for $10,000$ iterations. To learn the component weights, we apply the Gumble-Softmax trick (Jang et al., 2017; Maddison et al., 2017) with a temperature of $0.1$. Figure 1 shows the result when we obtain random mixtures $p$ using $s = 5$, when the dimension $d$ of $x$ equals 2 and 10, respectively.

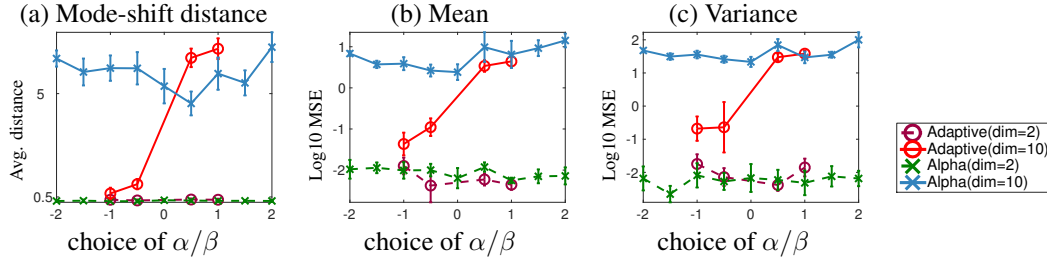

Figure 1: (a) plots the mode-shift distance between $p$ and $q$; (b-c) show the MSE of mean and variance between the true posterior $p$ and our approximation $q$, respectively. All results are averaged over 10 random trials.

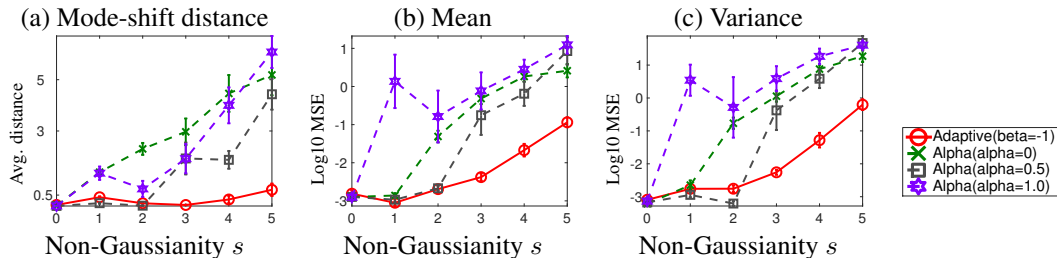

Figure 2: Results on randomly generated Gaussian mixture models. (a) plots the average mode-shift distance; (b-c) show the MSE of mean and variance. All results are averaged over 10 random trials.

We can see that when the dimension is low ($= 2$), all algorithms perform similarly well. However, as we increase the dimension to 10, our approach with tail-adaptive $f$-divergence achieves the best performance.

To examine the performance of variational approximation more closely, we show in Figure 2 the average mode-shift distance and the MSE of the estimated mean and variance as we gradually increase the non-Gaussianality of $p(x)$ by changing $s$. We fix the dimension to 10. We can see from Figure 2 that when $p$ is close to Gaussian (small $s$), all algorithms perform well; when $p$ is highly non-Gaussian (large $s$), we find that our approach with adaptive weights significantly outperform other baselines.

## 6.2 Bayesian Neural Network

We evaluate our approach on Bayesian neural network regression tasks. The datasets are collected from the UCI dataset repository[3]. Similarly to Li & Turner (2016), we use a single-layer neural network with 50 hidden units and ReLU activation, except that we take 100 hidden units for the Protein and Year dataset which are relatively large. We use a fully factorized Gaussian approximation to the true posterior and Gaussian prior for neural network weights. All datasets are randomly partitioned into 90% for training and 10% for testing. We use Adam optimizer (Kingma & Ba, 2015) with a constant learning rate of 0.001. The gradient is approximated by $n = 100$ draws of $x_i \sim q_\theta$ and a minibatch of size 32 from the training data points. All results are averaged over 20 random partitions, except for Protein and Year, on which 5 trials are repeated.

We summarize the average RMSE and test log-likelihood with standard error in Table 1. We compare our algorithm with $\alpha = 0$ (KL divergence) and $\alpha = 0.5$, which are reported as the best for this task in Li & Turner (2016). More comparisons with different choices of $\alpha$ are provided in the appendix. We can see from Table 1 that our approach takes advantage of an adaptive choice of $f$-divergence and achieves the best performance for both test RMSE and test log-likelihood in most of the cases.

|         | Average Test RMSE | | | Average Test Log-likelihood | | |
|---------|-----------------|-----------------|-----------------|-----------------|-----------------|-----------------|
| dataset | $\beta=-1.0$ | $\alpha=0.0$ | $\alpha=0.5$ | $\beta=-1.0$ | $\alpha=0.0$ | $\alpha=0.5$ |
| Boston | $\mathbf{2.828 \pm 0.177}$ | $2.956 \pm 0.171$ | $2.990 \pm 0.173$ | $\mathbf{-2.476 \pm 0.177}$ | $-2.547 \pm 0.171$ | $-2.506 \pm 0.173$ |
| Concrete | $\mathbf{5.371 \pm 0.115}$ | $5.592 \pm 0.124$ | $5.381 \pm 0.111$ | $\mathbf{-3.099 \pm 0.115}$ | $-3.149 \pm 0.124$ | $-3.103 \pm 0.111$ |
| Energy | $\mathbf{1.377 \pm 0.034}$ | $1.431 \pm 0.029$ | $1.531 \pm 0.047$ | $\mathbf{-1.758 \pm 0.034}$ | $-1.795 \pm 0.029$ | $-1.854 \pm 0.047$ |
| Kin8nm | $0.085 \pm 0.001$ | $0.088 \pm 0.001$ | $\mathbf{0.083 \pm 0.001}$ | $1.055 \pm 0.001$ | $1.012 \pm 0.001$ | $\mathbf{1.080 \pm 0.001}$ |
| Naval | $\mathbf{0.001 \pm 0.000}$ | $\mathbf{0.001 \pm 0.000}$ | $0.004 \pm 0.000$ | $5.468 \pm 0.000$ | $5.269 \pm 0.000$ | $4.086 \pm 0.000$ |
| Combined | $\mathbf{4.116 \pm 0.032}$ | $4.161 \pm 0.034$ | $4.154 \pm 0.042$ | $\mathbf{-2.835 \pm 0.032}$ | $-2.845 \pm 0.034$ | $-2.843 \pm 0.042$ |
| Wine | $0.636 \pm 0.008$ | $\mathbf{0.634 \pm 0.007}$ | $\mathbf{0.634 \pm 0.008}$ | $-0.962 \pm 0.008$ | $\mathbf{-0.959 \pm 0.007}$ | $-0.971 \pm 0.008$ |
| Yacht | $\mathbf{0.849 \pm 0.059}$ | $0.861 \pm 0.056$ | $1.146 \pm 0.092$ | $\mathbf{-1.711 \pm 0.059}$ | $-1.751 \pm 0.056$ | $-1.875 \pm 0.092$ |
| Protein | $\mathbf{4.487 \pm 0.019}$ | $4.565 \pm 0.026$ | $4.564 \pm 0.040$ | $\mathbf{-2.921 \pm 0.019}$ | $-2.938 \pm 0.026$ | $-2.928 \pm 0.040$ |
| Year | $\mathbf{8.831 \pm 0.037}$ | $8.859 \pm 0.036$ | $8.985 \pm 0.042$ | $-3.570 \pm 0.037$ | $-3.600 \pm 0.036$ | $\mathbf{-3.518 \pm 0.042}$ |

Table 1: Average test RMSE and log-likelihood for Bayesian neural regression.

## 6.3 Application in Reinforcement Learning

We now demonstrate an application of our method in reinforcement learning, applying it as an inner loop to improve a recent soft actor-critic(SAC) algorithm (Haarnoja et al., 2018). We start with a brief introduction of the background of SAC and then test our method in MuJoCo [4] environments.

**Reinforcement Learning Background**  Reinforcement learning considers the problem of finding optimal policies for agents that interact with uncertain environments to maximize the long-term cumulative reward. This is formally framed as a Markov decision process in which agents iteratively take actions $a$ based on observable states $s$, and receive a reward signal $r(s,a)$ immediately following the action $a$ performed at state $s$. The change of the states is governed by an unknown environmental dynamic defined by a transition probability $T(s'|s,a)$. The agent's action $a$ is selected by a conditional probability distribution $\pi(a|s)$ called policy. In policy gradient methods, we consider a set of candidate policies $\pi_\theta(a|s)$ parameterized by $\theta$ and obtain the optimal policy by maximizing the expected cumulative reward

$$J(\theta) = \mathbb{E}_{s \sim d_\pi, a \sim \pi(a|s)}\left[r(s,a)\right],$$

where $d_\pi(s) = \sum_{t=1}^\infty \gamma^{t-1}\Pr(s_t = s)$ is the unnormalized discounted state visitation distribution with discount factor $\gamma \in (0,1)$.

**Soft Actor-Critic**  (SAC) is an off-policy optimization algorithm derived based on maximizing the expected reward with an entropy regularization. It iteratively updates a Q-function $Q(a,s)$, which predicts that cumulative reward of taking action $a$ under state $s$, as well as a policy $\pi(a|s)$ which selects action $a$ to maximize the expected value of $Q(s,a)$. The update rule of $Q(s,a)$ is based on a variant of Q-learning that matches the Bellman equation, whose detail can be found in Haarnoja et al. (2018). At each iteration of SAC, the update of policy $\pi$ is achieved by minimizing KL divergence

$$\pi^{\text{new}} = \arg\min_\pi \mathbb{E}_{s \sim d}\left[\text{KL}(\pi(\cdot|s) \,||\, p_Q(\cdot|s))\right], \qquad (9)$$

$$p_Q(a|s) = \exp\left(\frac{1}{\tau}(Q(a,s) - V(s))\right), \qquad (10)$$

where $\tau$ is a temperature parameter, and $V(s) = \tau \log \int_a \exp(Q(a,s)/\tau)$, serving as a normalization constant here, is a soft-version of value function and is also iteratively updated in SAC. Here, $d(s)$ is a visitation distribution on states $s$, which is taken to be the empirical distribution of the states in the current replay buffer in SAC. We can see that (9) can be viewed as a variational inference problem on conditional distribution $p_Q(a|s)$, with the typical KL objective function ($\alpha = 0$).

**SAC With Tail-adaptive $f$-Divergence**  To apply $f$-divergence, we first rewrite (9) to transform the conditional distributions to joint distributions. We define joint distribution $p_Q(a,s) = \exp((Q(a,s) - V(s))/\tau)d(s)$ and $q_\pi(a,s) = \pi(a|s)d(s)$, then we can show that $\mathbb{E}_{s \sim d}[\text{KL}(\pi(\cdot|s) \,||\, p_Q(\cdot|s))] = \text{KL}(q_\pi \,||\, p_Q)$. This motivates us to extend the objective function in (9) to more general $f$-divergences,

$$D_f(p_Q \,||\, q_\pi) = \mathbb{E}_{s \sim d}\mathbb{E}_{a|s \sim \pi}\left[f\left(\frac{\exp((Q(a,s) - V(s))/\tau)}{\pi(a|s)}\right) - f(1)\right].$$

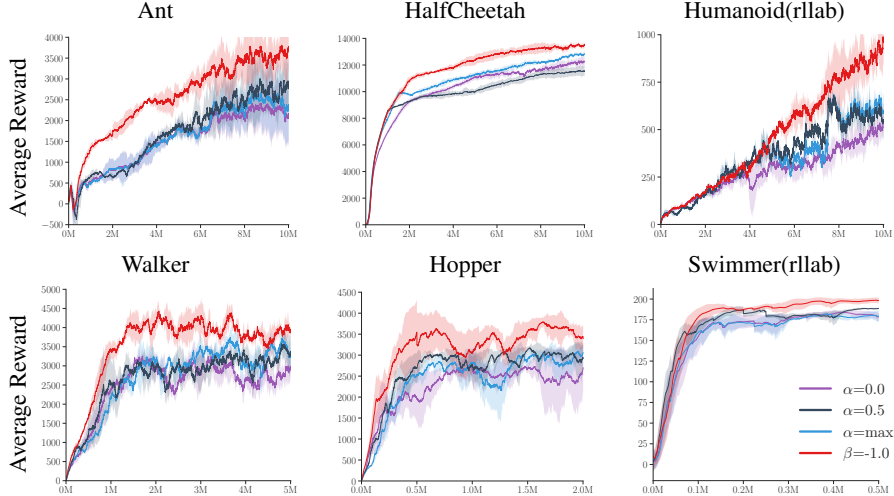

Figure 3: Soft Actor Critic (SAC) with policy updated by Algorithm 1 with $\beta = -1$, or $\alpha$-divergence VI with different $\alpha$ ($\alpha = 0$ corresponds to the original SAC). The reparameterization gradient estimator is used in all the cases. In the legend, "$\alpha = \max$" denotes setting $\alpha = +\infty$ in $\alpha$-divergence.

By using our tail-adaptive $f$-divergence, we can readily apply our Algorithm 1 (or Algorithm 2 in the Appendix) to update $\pi$ in SAC, allowing us to obtain $\pi$ that counts the multi-modality of $Q(a, s)$ more efficiently. Note that the standard $\alpha$-divergence with a fixed $\alpha$ also yields a new variant of SAC that is not yet studied in the literature.

**Empirical Results**  We follow the experimental setup of Haarnoja et al. (2018). The policy $\pi$, the value function $V(s)$, and the Q-function $Q(s, a)$ are neural networks with two fully-connected layers of 128 hidden units each. We use Adam (Kingma & Ba, 2015) with a constant learning rate of 0.0003 for optimization. The size of the replay buffer for HalfCheetah is $10^7$, and we fix the size to $10^6$ on other environments in a way similar to Haarnoja et al. (2018).

We compare with the original SAC ($\alpha = 0$), and also other $\alpha$-divergences, such as $\alpha = 0.5$ and $\alpha = \infty$ (the $\alpha = \max$ suggested in Li & Turner (2016)). Figure 3 summarizes the total average reward of evaluation rollouts during training on various MuJoCo environments. For non-negative $\alpha$ settings, methods with larger $\alpha$ give higher average reward than the original KL-based SAC in most of the cases. Overall, our adaptive $f$-divergence substantially outperforms all other $\alpha$-divergences on all of the benchmark tasks in terms of the final performance, and learns faster than all the baselines in most environments. We find that our improvement is especially significant on high dimensional and complex environments like Ant and Humanoid.

## 7   Conclusion

In this paper, we present a new class of tail-adaptive $f$-divergence and exploit its application in variational inference and reinforcement learning. Compared to classic $\alpha$-divergence, our approach guarantees finite moments of the density ratio and provides more stable importance weights and gradient estimates. Empirical results on Bayesian neural networks and reinforcement learning indicate that our approach outperforms standard $\alpha$-divergence, especially for high dimensional multi-modal distribution.

## Acknowledgement

This work is supported in part by NSF CRII 1830161. We would like to acknowledge Google Cloud for their support.

## Footnotes

*Work done at UT Austin

[2]Fat-tailed distributions is a sub-class of heavy-tailed distributions, which are distributions whose tail probabilities decay slower than exponential functions, that is, $\lim_{t \to +\infty} \exp(\lambda t) \bar{F}_w(t) = \infty$ for all $\lambda > 0$.

[3]`https://archive.ics.uci.edu/ml/datasets.html`

[4] http://www.mujoco.org/

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
