[Supplementary Material]

# A  Proof of Proposition 4.1

*Proof.* Taking $h(t) = f''(t)$, $a = f'(0)$ and $b = f(0)$ in Eq (5), we have

$$(f'(0)t + f(0)) \; + \; \int_0^\infty (t - \mu)_+ h(\mu) d\mu$$

$$= (f'(0)t + f(0)) \; + \; \int_0^t (t - \mu) f''(\mu) d\mu$$

$$= (f'(0)t + f(0)) \; + \; (t - \mu) f'(\mu) \Big|_{\mu=0}^t \; + \; \int_0^t f'(\mu) d\mu \quad \text{(integration by parts)}$$

$$= f(0) + \int_0^t f'(\mu) d\mu$$

$$= f(t).$$

Conversely, if $f(t) = (at + b) + \int_0^\infty (t - \mu)_+ h(\mu) d\mu$, calculation shows

$$f'(t) = a + \int_0^t h(\mu) d\mu, \quad f''(t) = h(t).$$

Therefore, $f$ is convex if $h$ is non-negative.

To prove Eq. (6), we substitute $f(t) = f'(0)t + f(0) \; + \; \int_0^\infty (t - \mu)_+ f''(\mu) d\mu$ into the definition of $f$-divergence,

$$D_f(p \,||\, q) = \mathbb{E}_q \left[ f\left( \frac{p(x)}{q(x)} \right) - f(1) \right]$$

$$= \mathbb{E}_q \left[ f'(0) \frac{p(x)}{q(x)} + f(0) \; + \; \int_0^\infty (p(x)/q(x) - \mu)_+ f''(\mu) d\mu - f(1) \right]$$

$$= [f'(0) + f(0) - f(1)] \; + \; \int_0^\infty \mathbb{E}_q \left[ \left( \frac{p(x)}{q(x)} - \mu \right)_+ \right] f''(\mu) d\mu.$$

This completes the proof. $\qquad\square$

# B  Proof of Proposition 4.2

*Proof.* By chain rule and the "score-function trick" $\nabla_\theta q_\theta(x) = q_\theta(x) \nabla_\theta \log q_\theta(x)$, we have

$$\nabla_\theta D_f(p \,||\, q_\theta) = \mathbb{E}_{q_\theta} \left[ \nabla_\theta f\left( \frac{p(x)}{q_\theta(x)} \right) + f\left( \frac{p(x)}{q_\theta(x)} \right) \nabla_\theta \log q_\theta(x) \right]$$

$$= \mathbb{E}_{q_\theta} \left[ f'\left( \frac{p(x)}{q_\theta(x)} \right) \nabla_\theta \left( \frac{p(x)}{q_\theta(x)} \right) + f\left( \frac{p(x)}{q_\theta(x)} \right) \nabla_\theta \log q_\theta(x) \right]$$

$$= \mathbb{E}_{q_\theta} \left[ -f'\left( \frac{p(x)}{q_\theta(x)} \right) \left( \frac{p(x)}{q_\theta(x)} \right) \nabla_\theta \log q_\theta(x) + f\left( \frac{p(x)}{q_\theta(x)} \right) \nabla_\theta \log q_\theta(x) \right]$$

$$= -\mathbb{E}_{q_\theta} \left[ \rho_f\left( \frac{p(x)}{q_\theta(x)} \right) \log q_\theta(x) \right],$$

where $\rho_f(t) = f'(t)t - f(t)$. This proves Eq. (7).

To prove Eq. (8), we note that for any function $\phi$, we have by the *reparamertization trick*:

$$\nabla_\theta \mathbb{E}_{q_\theta}[\phi(x)] = \mathbb{E}_{x \sim q_\theta}[\phi(x) \nabla_\theta \log q_\theta(x)] \quad \text{(score function)}$$

$$= \mathbb{E}_{\xi \sim q_0}[\nabla_x \phi(x) \nabla_\theta g_\theta(\xi)] \quad \text{(reparameterization trick)},$$

where we assume $x \sim q_\theta$ is generated by $x = g_\theta(\xi), \; \xi \sim q_0$.

Taking $\phi(x) = \rho_f(p(x)/q_\theta(x))$ in Eq. (7), we have

$$
\begin{aligned}
\nabla_\theta D_f(p \,||\, q_\theta) &= -\mathbb{E}_{x \sim q_\theta}\left[\rho_f\left(\frac{p(x)}{q_\theta(x)}\right)\nabla_\theta \log q_\theta(x)\right] \\
&= -\mathbb{E}_{\xi \sim q_0}\left[\nabla_x \rho_f\left(\frac{p(x)}{q_\theta(x)}\right)\nabla_\theta g_\theta(\xi)\right] \\
&= -\mathbb{E}_{\xi \sim q_0}\left[\rho_f'\left(\frac{p(x)}{q_\theta(x)}\right)\nabla_x\left(\frac{p(x)}{q_\theta(x)}\right)\nabla_\theta g_\theta(\xi)\right] \\
&= -\mathbb{E}_{\xi \sim q_0}\left[\rho_f'\left(\frac{p(x)}{q_\theta(x)}\right)\left(\frac{p(x)}{q_\theta(x)}\right)\nabla_x \log\left(\frac{p(x)}{q_\theta(x)}\right)\nabla_\theta g_\theta(\xi)\right] \\
&= -\mathbb{E}_{\xi \sim q_0}\left[\gamma_f\left(\frac{p(x)}{q_\theta(x)}\right)\nabla_x \log\left(\frac{p(x)}{q_\theta(x)}\right)\nabla_\theta g_\theta(\xi)\right],
\end{aligned}
$$

where $\gamma_f(t) = \rho_f'(t)t$.

$\square$

## C  Tail-adaptive $f$-divergence with Score-Function Gradient Estimator

Algorithm 2 summarizes our method using the score-function gradient estimator (7).

---

**Algorithm 2** Variational Inference with Tail-adaptive $f$-Divergence (with Score Function Gradient)

---

Goal: Find the best approximation of $p(x)$ from $\{q_\theta : \theta \in \Theta\}$.
Initialize $\theta$, set an index $\beta$ (e.g., $\beta = -1$).
**for** iteration **do**
    Draw $\{x_i\}_{i=1}^n \sim q_\theta$. Set $\hat{\tilde{F}}(t) = \sum_{j=1}^n \mathbb{I}(p(x_j)/q(x_j) \geq t)/n$, and $\rho_i = \hat{\tilde{F}}(p(x_i)/q(x_i))^\beta$.
    Update $\theta \leftarrow \theta + \epsilon\Delta\theta$, where $\epsilon$ is stepsize, and

$$
\Delta\theta = \frac{1}{z_\rho}\sum_{i=1}^n \left[\rho_i \nabla_\theta \log q_\theta(x_i)\right],
$$

    where $z_\rho = \sum_{i=1}^n \rho_i$.
**end for**

---

## D More Results for Bayesian Neural Network

Table 2 shows more results in Bayesian networks with more choices of $\alpha$ in $\alpha$-divergence. We can see that our approach achieves the best performance in most of the cases.

| | Average Test RMSE | | | | | | | |
|---|---|---|---|---|---|---|---|---|
| Dataset | $\beta = -1.0$ | $\beta = -0.5$ | $\alpha = -1$ | $\alpha = 0$ | $\alpha = 0.5$ | $\alpha = 1.0$ | $\alpha = 2.0$ | $\alpha = +\infty$ |
| Boston | **2.828** | 2.948 | 3.026 | 2.956 | 2.990 | 2.937 | 2.981 | 2.985 |
| Concrete | **5.371** | 5.505 | 5.717 | 5.592 | 5.381 | 5.462 | 5.499 | 5.481 |
| Energy | **1.377** | 1.461 | 1.646 | 1.431 | 1.531 | 1.413 | 1.458 | 1.458 |
| Kin8nm | 0.085 | 0.088 | 0.087 | 0.088 | **0.083** | 0.084 | 0.084 | **0.083** |
| Naval | **0.001** | **0.001** | 0.003 | **0.001** | 0.004 | 0.005 | 0.004 | 0.004 |
| Combined | **4.116** | 4.146 | 4.156 | 4.161 | 4.154 | 4.135 | 4.188 | 4.145 |
| Wine | 0.636 | **0.632** | **0.632** | 0.634 | 0.634 | 0.633 | 0.635 | 0.634 |
| Yacht | 0.849 | **0.788** | 1.478 | 0.861 | 1.146 | 1.221 | 1.222 | 1.234 |
| Protein | **4.487** | 4.531 | 4.550 | 4.565 | 4.564 | 4.658 | 4.777 | 4.579 |
| Year | **8.831** | 8.839 | 8.841 | 8.859 | 8.985 | 9.160 | 9.028 | 9.086 |
| | Average Test Log-likelihood | | | | | | | |
| dataset | $\beta = -1.0$ | $\beta = -0.5$ | $\alpha = -1$ | $\alpha = 0$ | $\alpha = 0.5$ | $\alpha = 1.0$ | $\alpha = 2.0$ | $\alpha = +\infty$ |
| Boston | **-2.476** | -2.523 | -2.561 | -2.547 | -2.506 | -2.493 | -2.516 | -2.509 |
| Concrete | **-3.099** | -3.133 | -3.171 | -3.149 | -3.103 | -3.106 | -3.116 | -3.109 |
| Energy | **-1.758** | -1.814 | -1.946 | -1.795 | -1.854 | -1.801 | -1.828 | -1.832 |
| Kin8nm | 1.055 | 1.017 | 1.024 | 1.012 | 1.080 | 1.075 | 1.074 | **1.085** |
| Naval | **5.468** | 5.347 | 4.178 | 5.269 | 4.086 | 4.022 | 4.077 | 4.037 |
| Combined | **-2.835** | -2.842 | -2.845 | -2.845 | -2.843 | -2.839 | -2.850 | -2.842 |
| Wine | -0.962 | **-0.956** | -0.961 | -0.959 | -0.971 | -0.968 | -0.972 | -0.971 |
| Yacht | **-1.711** | -1.718 | -2.201 | -1.751 | -1.875 | -1.946 | -1.963 | -1.986 |
| Protein | **-2.921** | -2.930 | -2.934 | -2.938 | -2.928 | -2.930 | -2.947 | -2.932 |
| Year | -3.570 | -3.597 | -3.599 | -3.600 | **-3.518** | -3.529 | -3.524 | -3.524 |

Table 2: Test RMSE and LL results for Bayesian neural network regression.

## E Reinforcement Learning

In this section, we provide more information and results of the Reinforcement learning experiments, including comparisons of algorithms using score-function gradient estimators (Algorithm 2).

### E.1 MuJoCo Environments

We test six MuJoCo environments in this paper: *HalfCheetah*, *Hopper*, *Swimmer(rllab)*, *Humanoid(rllab)*, *Walker*, and *Ant*, for which the dimensions of the action space are 6, 3, 2, 21, 6, 8, respectively. Figure 4 shows examples of the environment used in our experiments.

Figure 4: MuJoCo environments used in our reinforcement learning experiments. From left to right: HalfCheetah, Hopper, Swimmer(rllab), Humanoid(rllab), Walker, and Ant.

### E.2 Different Choices of $\alpha$

In this section, we present the average reward of $\alpha$-divergences with different choices of $\alpha$ on Hopper and Walker with both score-function and reparameterization gradient estimators. We can see from Figure 5 that $\alpha = 0.5$ and $\alpha = +\infty$ (denoted by $\alpha = \max$ in the legends) perform consistently better than standard KL divergence ($\alpha = 0$), which is used the original SAC paper.

Figure 5: Results on Hopper and Walker with different choices of $\alpha$.

### E.3 Tail-adaptive $f$-divergence with score function estimation

In this section, we investigate optimization with score function gradient estimators (Algorithm 2). The results in Figure 6 show that our tail-adaptive $f$-divergence tends to yield better performance across all environments tested.

Figure 6: Results of average rewards with score function gradients.