[Reviews · NeurIPS 2018]

Reviewer 1



# Paper ID 2274 Variational Inference with Tail Adapted f-divergence - The paper proposes an improved variational inference method, f-divergence method that improves over alpha divergence (and KL divergence). The paper is well written and the problem considered is extremely relevant to NIPS. The ideas are theoretically justified and the approximations made are principled. The experiments are well designed and show good performance compared to alpha divergence. - The method seeks to improve upon alpha divergence. Specifically, the alpha parameter, kept constant during trianing, may experience problems associated with fat-tailed distributions due to variation in the density ratio. Consequently, adjusting alpha is difficult. A constant alpha is the current state-of-the-art which this algorithm improves upon. - The main contribution is a theoretically justified method to compute the gradient of the f divergence without knowing f itself. Both variants (score function, reparameterization) are described. After making computational approximations, the resulting algorithm is a stochastic approximation to the gradient of the f-divergence. Computationally, the key simplification to the approximation is using the (non-negative) rank of the density ratio to weight each data point in the sample. - The experimental section is compelling. The results demonstrate the practical utility of the ideas and approximations. In particular, the method outperforms the previous state-of-the-art (alpha divergence) on the hardest instances. This aligns nicely with the theory making for a very nice result. The same trend occurs in all three problem types considered (mixture model, regression task, SAC). - As far as I can tell, this paper proposes a novel variational inference algorithm with strong theoretical justifications, improving upon the previous best method. The experiments clearly show the superiority of the proposed method. The paper is very well written (with a few minor typos) and seems like an novel contribution to the variational inference literature.

Reviewer 2



This paper proposes an adaptive f-divergence for variational inference (VI), that both guarantees that importance weights mean is finite, and reduces fat-tail distributions. The approach entails representing f-divergence as a Hessian enforce its convexity, and using rank of the importance weights at each gradient descent of posterior approximation. In the process of evaluation, the authors provide an improved variant of the soft actor critic reinforcement learning algorithm. The paper is exceptionally well motivated and placed within the related work. The concepts are introduced clearly. The new method is elegant, results in a simple algorithm that lies on top of in-depth theoretical analysis. The key insight is a clever representation of function as a Hessian. The results are promising and convincing. And that the paper contains additional reinforcement learning algorithm as a side effect is quite impressive. This work is very significant and original. The only concern with the paper is the lack of proofs for the propositions 4.1.a and 5.1. Given the convincing results this does not invalidate the paper, although it would make it stronger. Also, please make a note that the proofs for Propositions 4.2 and 4.3 are in the appendix.

Reviewer 3



When approximating alpha divergence from samples, the estimation variance can be very high due to the unbounded density ratio. In this paper, a new variation of f-divergence is introduced. It guarantees a finite mean of importance weights and its reparametrization gradient can be easily calculated from empirical samples. Experiments on variational inference and reinforcement learning show promising results. This is a very interesting paper which addresses an important question: in many density ratio-based divergence estimation (such as KL divergence), the fat-tail property of density ratio causes large estimation variance. The alternative definition of the f-divergence shows an interesting way of avoiding such a problem as well as computing the empirical gradient of the divergence minimization problem. The use of emperical CDF of density ratio for suppressing the expectation of f(ratio) is interesting but I still have some concerns: Proposition 5.1, authors proved the expectation of phi(ratio) is bounded: 1. With respect to what distribution, the expectation is taken? p or q? 2. The proposition only states the bounded mean. Since authors talked about estimation variance of alpha divergence, I thought the second order moment of phi(ratio) might be more interesting than the mean? 3. What is the estimation variance of the gradient in algorithm 1? How does it compare with the alpha divergence? Otherwise, this paper is well-written and easy to understand. I would like to recommend for acceptance. ------------------------------------------------------------------------------------------------------------- I have read the feedback from authors. I would upgrade my score by one after the clarification.